# Chronic Kidney Disease and Nephrology Care in People Living with HIV in Central/Eastern Europe and Neighbouring Countries—Cross-Sectional Analysis from the ECEE Network

**DOI:** 10.3390/ijerph191912554

**Published:** 2022-10-01

**Authors:** Bartłomiej Matłosz, Agata Skrzat-Klapaczyńska, Sergii Antoniak, Tatevik Balayan, Josip Begovac, Gordana Dragovic, Denis Gusev, Djordje Jevtovic, David Jilich, Kerstin Aimla, Botond Lakatos, Raimonda Matulionyte, Aleksandr Panteleev, Antonios Papadopoulos, Nino Rukhadze, Dalibor Sedláček, Milena Stevanovic, Anna Vassilenko, Antonija Verhaz, Nina Yancheva, Oleg Yurin, Andrzej Horban, Justyna D. Kowalska

**Affiliations:** 1HIV Outpatient Clinic, Hospital for Infectious Diseases, Medical University of Warsaw, 02-091 Warszawa, Poland; 2Department of Adults’ Infectious Diseases, Hospital for Infectious Diseases, Medical University of Warsaw, 02-091 Warszawa, Poland; 3Viral Hepatitis and AIDS Department, Gromashevsky Institute of Epidemiology and Infectious Diseases, 01001 Kyiv, Ukraine; 4National Center for Disease Control and Prevention, Yerevan 0002, Armenia; 5School of Medicine, University Hospital for Infectious Diseases, University of Zagreb, 10000 Zagreb, Croatia; 6Department of Pharmacology, Clinical Pharmacology and Toxicology, School of Medicine, University of Belgrade, 11000 Belgrade, Serbia; 7Botkin’s Infectious Disease Hospital, First Saint-Petersburg State Medical University Named after I.P. Pavlov, 197022 Saint-Petersburg, Russia; 8Infectious Disease Hospital, Belgrade University School of Medicine, 11000 Belgrade, Serbia; 9Department of Infectious Diseases, 1st Faculty of Medicine, Charles University in Prague and Faculty Hospital Bulovka Hospital, 18000 Prague, Czech Republic; 10West Tallinn Central Hospital, 10111 Tallinn, Estonia; 11National Institute of Hematology and Infectious Diseases, South-Pest Central Hospital, National Center of HIV, 1007 Budapest, Hungary; 12Faculty of Medicine, Vilnius University, Vilnius University Hospital Santaros Klinikos, 08410 Vilnius, Lithuania; 13City TB Dispensary, 101000 Moscow, Russia; 14University General Hospital Attikon, Medical School, National and Kapodistrian University of Athens, 15772 Athens, Greece; 15Infectious Diseases, AIDS and Clinical Immunology Center, 112482 Tblisi, Georgia; 16Faculty of Medicine in Plzeň, University Hospital Plzeň, Charles University, 30599 Plzen, Czech Republic; 17University Clinic for Infectious Diseases and Febrile Conditions, 1000 Skopje, North Macedonia; 18Global Fund Grant Management Department, Republican Scientific and Practical Center for Medical Technologies, 220004 Minsk, Belarus; 19Department for Infectious Diseases, Faculty of Medicine, University of Banja Luka, 78 000 Banja Luka, Republika Srpska, Bosnia and Herzegovina; 20Department for AIDS, Specialized Hospital for Active Treatment of Infectious and Parasitic Disease, 1000 Sofia, Bulgaria; 21Central Research Institute of Epidemiology, Federal AIDS Centre, 101000 Moscow, Russia

**Keywords:** HIV, chronic kidney disease, Central and Eastern Europe

## Abstract

Chronic kidney disease (CKD) is a significant cause of morbidity and mortality among patients infected with human immunodeficiency virus (HIV). The Central and East Europe (CEE) region consists of countries with highly diversified HIV epidemics, health care systems and socioeconomic status. The aim of the present study was to describe variations in CKD burden and care between countries. The Euroguidelines in the CEE Network Group includes 19 countries and was initiated to improve the standard of care for HIV infection in the region. Information on kidney care in HIV-positive patients was collected through online surveys sent to all members of the Network Group. Almost all centres use regular screening for CKD in all HIV (+) patients. Basic diagnostic tests for kidney function are available in the majority of centres. The most commonly used method for eGFR calculation is the Cockcroft–Gault equation. Nephrology consultation is available in all centres. The median frequency of CKD was 5% and the main cause was comorbidity. Haemodialysis was the only modality of treatment for kidney failure available in all ECEE countries. Only 39% of centres declared that all treatment options are available for HIV+ patients. The most commonly indicated barrier in kidney care was patients’ noncompliance. In the CEE region, people living with HIV have full access to screening for kidney disease but there are important limitations in treatment. The choice of dialysis modality and access to kidney transplantation are limited. The main burden of kidney disease is unrelated to HIV infection. Patient care can be significantly improved by addressing noncompliance.

## 1. Introduction

Chronic kidney disease is a significant cause of morbidity and mortality among patients infected with the human immunodeficiency virus (HIV). The prevalence around the world in the HIV-infected population varies but, in most reports, it is estimated to be between 4.7% and 9.7% [1,2,3,4,5]. With effective antiretroviral therapy (ARV), life expectancy in individuals with HIV has increased. As a consequence, the spectrum of kidney diseases in people living with HIV has broadened, including not only HIV-related problems or drug toxicity but also renal damage from chronic noncommunicable diseases.

The Central and East Europe (CEE) region with a population of about 300 million consists of several countries with highly diversified HIV epidemics, health care systems and socio-economic status [6]. The data on kidney disease and health care in the region used to be scarce. In many reports regarding kidney disease prevalence and kidney care, the countries from the region are labelled as ‘no data available’ [5,7]. The availability of the data from the region is, however, improving and in the latest issue of the International Society of Nephrology Global Kidney Health Atlas, many of the blind spots were filled with data [8]. Still, the lack of national registries in the region remains a substantial obstacle to collecting reliable and comparable data. The ISN Global Kidney Health Atlas indicates 18 regional registries as the data source for Western Europe (out of 25 countries) and only 2 regional registries for the Central and Eastern Europe region (out of 19 countries). The resources available for kidney care seem to improve across the CEE region, although they still tend to lag behind other parts of Europe, especially in terms of low or very low rates of kidney transplantation [9].

As the data on renal disease in the population of people living with HIV are even more limited than in the general population, the present study aimed to investigate the state and limitations of care for chronic kidney disease and end-stage kidney disease in countries represented in the Euroguidelines in Central and Eastern Europe (ECEE) Network Group.

## 2. Materials and Methods

Euroguidelines in Central and Eastern Europe Network Group was initiated in February 2016 to compare and improve the standard of care for HIV infection in the region. Information on kidney care in HIV-positive patients was collected through online surveys sent to all members of the ECEE Network Group in November 2018. Respondents were ECEE members from 20 countries in the region (Albania, Armenia, Belarus, Bosnia and Herzegovina, Bulgaria, Croatia, Czech Republic, Estonia, Georgia, Greece, Hungary, Lithuania, Macedonia, Poland, Republic of Moldova, Romania, Russia, Serbia, Slovenia, Turkey and Ukraine). 

The collected data were exported to the R statistical software. All analyses were performed using R software (version 3.6.2). The responses were based on real-world data, including centres’ own databases. As all of the responders were practitioners actively involved in patient care, data regarding the availability of diagnostic tools and treatment relayed on their real experience. Some answers had to be analysed by country (e.g., coverage from public funding). In such cases, data received from the same country were used for validation purposes and then aggregated for by-country analysis. 

Survey questions regarded institutional data, different aspects of nephrology care screening and diagnostic tests for kidney disease, availability of specialized nephrology care, burden and causes of kidney disease and guidelines employment. All survey questions can be found in Appendix A. The study did not include individual patient data and did not require ethical committee approval.

## 3. Results

The survey was sent to 43 members of the ECEE Network in 20 countries. We received 18 responses from 16 countries. According to the World Bank classification, among the responding countries, there were three lower–middle-income countries, five upper–middle-income countries and seven high–income countries. The vast majority of responding centres were described as treating PLWH as well as other infectious diseases (61%) and both hospital and outpatient-based (83%). The median population treated in a single centre was 1415 (IQR 600–2514) patients.

In most ECEE countries, the population of PLWH is relatively young with only four centres where patients > 65 years account for at least 10% of the population. In nine centres (50%), the proportion of elderly patients is less than 5%. We did not observe any significant correlation between age and the frequency of CKD. MSM was the most common route of HIV infection in 10 centres (55.5%). Insix6 centres (33.3%), IVDU is a common route of infection (more than 25% of infected patients). In 11 centres (61.1%), the majority of patients receive ARV therapy within one year after HIV infection diagnosis.

All but one centre use regular screening for kidney disease in all HIV (+) patients irrespective of ARV therapy usually within 3–6 month intervals. Among basic diagnostic tests, creatinine and abdominal ultrasound are available in all centres. In a few centres, urinalysis and albumin-to-creatinine ratio are not available (11.1%, *n* = 2 and 27.8%, *n* = 5, respectively). NMR, biopsy, scintigraphy and cystatin are rarely available (less than 50% of centres). Only a few centres did not use eGFR (22%, *n* = 4) nor albumin (28%, *n* = 5) for chronic kidney disease screening (Figure 1). The most commonly used method for eGFR calculation is still the Cockcroft–Gault Equation, but the CKD-EPI Equation is used with a similar frequency (38.9%, *n* = 7 and 33.3%, *n* = 6, respectively). MDRD formula is used only in three centres (16.7%). In four centres (22.2%), the eGFR method is not known. 

Specialized nephrology consultation is available in all centres. In three centres, a nephrologist is available on-site on a regular basis. In nine centres, consultation is available on call, and in five centres, patients are referred for consultation to the external institution. Nevertheless, the waiting time for consultation in any of the centres is not longer than 1 month and in half of the cases, it is up to one week. In the majority of centres, nephrology care is provided without any fee, although in seven centres, patients have to partly contribute to costs.

The main causes of chronic kidney disease are not related to HIV infection. The majority of responders (55%) answered that the first most common cause of chronic kidney disease is comorbidity (i.e., hypertension and diabetes). As the second most common causes of CKD, antiretrovirals’ and other drugs’ nephrotoxicity were indicated most commonly (33% and 28% responses, respectively). The most common and second most common causes of chronic kidney disease are depicted in (Figure 2A,B).

The median frequency of CKD was 5%, although there was significant variability between countries ranging from 0% to 20% of HIV-positive patients. With 37 reported cases out of over 58,000 individuals in care, end-stage renal disease is rare (0.06%) and only a few patients require RRT in most centres. Additionally, death resulting from end-stage renal disease is rare with no or single cases reported from almost all centres.

The only modality of treatment for ESRD completely covered from public funding (no matter of HIV status) in all ECEE countries is haemodialysis. Respondents from four countries declared that neither living nor deceased donor transplantation is founded on public funds. In five countries, the treatment of CKD complications and peritoneal dialysis is not covered. Only seven centres (38.9%) declared that all treatment options for ESRD (haemodialysis, peritoneal dialysis, and kidney transplantation) are available for HIV+ patients. In 11 centres (61.1%), the only treatment option was dialysis, and among them, in 6 centres (33.3%), only haemodialysis was possible.

In most centres, treatment of chronic kidney disease is the responsibility of a nephrologist or interdisciplinary team (72.2%). In five centres (27.8%), infectious disease specialists are primarily responsible for kidney care in people living with HIV. The most commonly indicated barrier in kidney care was patients’ noncompliance (six centres). Other issues (i.e., health service availability, nephrologist availability, and distance from care point) were also common.

The most commonly used guidelines were EACS guidelines: eight centres (44.4%) use only EACS and seven centres (38.9%) both EACS and national ones. Guidelines adoption was usually described as moderate (61%) centres.

## 4. Discussion

The reported prevalence of CKD varied from 0 to 20% (median 5%). The highest rates (more than 10%) were observed in Hungary, Poland and Serbia. A very low prevalence of CKD (below 2%) was observed in Macedonia, Georgia, Estonia and Lithuania. In the general population, the prevalence of chronic kidney disease worldwide is 1.5–21% and in fact, it is highly variable in seemingly similar countries such as in Europe [7,10]. Significant differences were observed even within one country [11]. Aumann et al. showed that in different regions in Germany, the prevalence may be higher by a factor of 2. In the Pomerania region (SHIP-1 study) in Northeast Germany and the region of Augsburg in Southern Germany’s Cooperative Health Research Study (KORA F4), the prevalence was 5.9% and 3.1% accordingly. The same is true for HIV-positive patients in Europe [2,12]. The population of people living with HIV across the ECEE countries network is widely variable. Our results show huge disparities in epidemics across quite a small geographical region. There are countries with a high prevalence of patients infected through intravenous drug use as well as countries with MSM population dominance. The prevalence of elderly patients in the HIV-infected population may vary from as high as 20% in some countries to almost no such patients, as reported in four countries where less than 1% of patients are aged 65 or more. This may result in different risk factors for chronic kidney disease in different countries. Although many authors attribute rising chronic kidney disease to the ageing HIV (+) population, we were not able to see any significant increase in the reported CKD prevalence in centres with a high proportion of elderly patients [13,14].

According to EACS guidelines, screening for renal disease (eGFR and urine protein) should be carried out in every HIV-positive patient at least once a year (with a wide range of every 3–12 months) [15]. In patients with risk factors for established CKD, the frequency of eGFR monitoring should be increased. Additionally, in patients with decreased eGFR or proteinuria, abdominal ultrasound should be performed. In this survey regarding kidney care clinical practice in people living with HIV, we observed general good availability of diagnostic tests and treatment in chronic kidney disease. Most of the kidney function tests were available in all centres and specialized nephrology care was provided without delay. It is worth mentioning that some centres still do not utilize urinary albumin/protein for screening for kidney disease which is required by EACS guidelines [15]. Poorer availability of some specialized diagnostic tests such as nuclear magnetic resonance imaging or kidney scintigraphy should not be a problem as almost all centres reported the possibility of patients’ referral to a nephrologist for specialized care.

The definition of chronic kidney disease is based on the widely accepted KDIGO classification of CKD [16]. For GFR estimation, EACS guidelines recommend the use of the CKD-EPI formula, although it indicates that the abbreviated Modification of Diet in Renal Disease (MDRD) or the Cockcroft–Gault (CG) equation may be used as an alternative. KDIGO guidelines state that eGFR should be calculated with the CKD-EPI formula and an alternative creatinine-based GFR-estimating equation is acceptable only if it has been shown to improve the accuracy of GFR estimates compared to the 2009 CKD-EPI creatinine equation [16]. Almost all centres employ regular universal screening for kidney injury at 3–6 month intervals, which is even more frequent than required by EACS guidelines. The vast majority of the centres (67%) still use, for the estimation of GFR, equations other than the primarily recommended CKD-EPI equation. This fact may potentially have clinical significance as the CKD-EPI creatinine formula was validated in the HIV (+) population in many studies and seems to outperform other formulas by means of accuracy [17,18].

HIV may cause kidney injury in several ways. The classic kidney disease of HIV infection, HIV-associated nephropathy (HIVAN), causes rapid kidney function deterioration and, if not treated, usually leads to end-stage renal disease. The incidence of HIV-associated nephropathy decreased with the use of highly active antiretroviral therapy [19]. Additionally, a number of immune complex kidney diseases have been reported in patients with HIV infection, including membranous nephropathy, membranoproliferative and mesangial proliferative glomerulonephritis, and “lupus-like” proliferative glomerulonephritis [20,21]. However, the ageing cohort of HIV-positive patients may be at increased risk for kidney disease unrelated to direct HIV injury. Coinfections and comorbid or treatment-related diabetes and hypertension may play an important role [22]. The increasing role of traditional risk factors for CKD seems to be supported by the results of our survey. More than half of the centres indicated that comorbidity was the most common cause of CKD. Other commonly reported causes were nephrotoxicity of drugs and illegal substances. Only one centre reported HIV nephropathy as the main cause of chronic kidney disease. As the second most common causes of chronic kidney disease, ARV nephrotoxicity and other drugs’ or illegal substances’ nephrotoxicity were reported in the majority of centres.

Even in centres with a high prevalence of chronic kidney disease, end-stage renal disease was not a common problem. Only single cases of end-stage renal disease were reported. Only one country reported a significant number of patients requiring renal replacement therapy but not treated. The possible reasons include not only resource shortage but also patients’ noncompliance indicated by many participating centres as an important obstacle in providing care. Noncompliance was also described in previous studies as a serious problem with more than 50% of patients not attending the scheduled consultations [23]. No country reported differences in kidney care for HIV and the general population. Some centres indicated that kidney transplantation is founded on the public health insurance system but there is no possibility for HIV (+) individuals for transplantation. This may indicate that access to care, in reality, is not truly equal. Another possible explanation is that in many ECEE countries, kidney transplant programs have a very limited capacity [24]. Thus, the availability of transplantation may be limited in general in those countries despite public funding.

In general, access to nephrology care for people living with HIV is seemingly good. All centres employ regular renal function screening and in the vast majority of centres, nephrology consultation is possible without a delay. Nevertheless, our findings showed the lack of treatment options for ESRD in HIV-positive patients in a substantial proportion of Central and Eastern Europe Countries. In the ECEE region, one country in four has no public funding for kidney transplantation (which is also true for HIV-negative individuals). The only renal replacement therapy reimbursed in all ECEE countries is haemodialysis. The fact that treatment of CKD complications is also not covered by public health insurance in many countries may negatively influence the patients’ prognosis. Additionally, in many centres, chronic kidney disease treatment is solely the responsibility of the infectious disease specialists which is not optimal and is against current guidelines.

All centres indicate that there are barriers to kidney care access for HIV patients. The most commonly indicated obstacle was patients’ noncompliance, although all the other answers (distance from care, nephrologist availability, and healthcare system access) were also commonly indicated.

There are some important limitations to be discussed. This was an online survey-based study where we preselected respondents based on our best knowledge of expertise and up-to-date acquaintance with epidemiological and clinical data in their centres. Secondly, the source of information on coinfection prevalence varied from personal communication to detailed epidemiological surveillance; thus, the weight of the data presented may vary significantly across countries.

## 5. Conclusions

In the ECEE region, people living with HIV have full access to screening for kidney disease. The screening might be improved by the employment of albuminuria screening in all centres. There are some important limitations in access to renal replacement therapy both regarding the choice of dialysis modality and kidney transplantation. It must be stated that those limitations are also true for the HIV (−) population. The main burden of kidney disease in the ECEE region is not directly related to HIV infection and treatment but comorbidity and patient care can be improved by addressing noncompliance.

## Figures and Tables

**Figure 1 ijerph-19-12554-f001:**
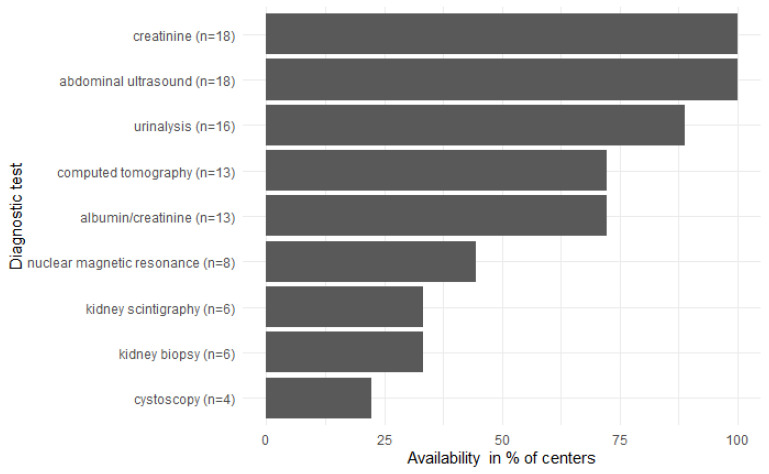
Availability of diagnostic tests.

**Figure 2 ijerph-19-12554-f002:**
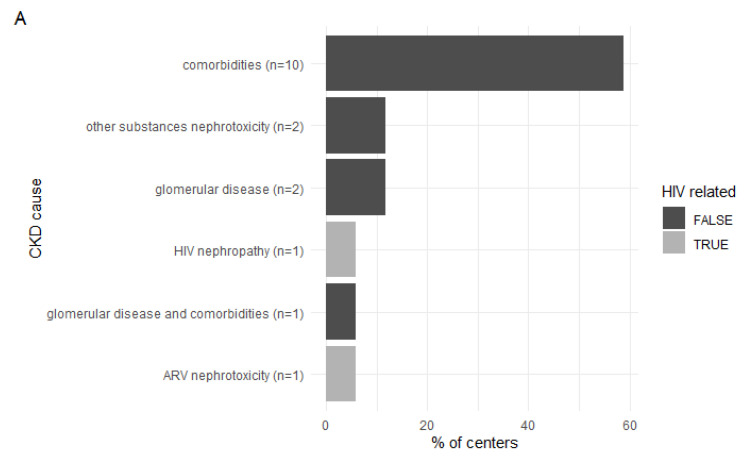
(**A**) The most common causes of CKD. (**B**) The second most common causes of CKD.

## Data Availability

Not applicable.

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
