# Peer review of "Chronic Kidney Disease and Nephrology Care in People Living with HIV in Central/Eastern Europe and Neighbouring Countries—Cross-Sectional Analysis from the ECEE Network"

_ijerph, 2022, doi:10.3390/ijerph191912554_

Round 1
Reviewer 1 Report
The manuscript “Chronic renal disease and nephrology care in people living 2 with HIV in Central/Eastern Europe and neighbors countries - data from the ECEE network” brings interesting information about nephrology care in people living with HIV, but lack of information about the methodology and ethical issues that are mandatory in a scientific publication. Therefore, a major revision of the text is necessary for a better understanding of the study:
1 - Describe the type of study in the summary and methodology
2 - Include information about the data collection and analysis, including the period of data collection, database construction, analysis description and software used
3 - Include information about ethical consideration
Author Response
Thank you for all the comments and remarks. We addressed all of them in the corrected manuscript. To summarise:
- we added a section in "Methods" explaining the methodology issues rised in points 1 and 2
- as this was a survey among clinicians, ethics comeete decision was not required. We make such a declaration in submission process but to clarify things we also included such a steatment in the manuscript.
Reviewer 2 Report
The authors present a detailed review of an important topic. A few questions. Line 104: it appears that only 18 of 43 members responded? How many countries did not have data included because their members did not respond? Were there any trends that could have impacted the author's conclusions (i. e. were countries with larger or smaller populations more or less likely to be included in the data analysis?) The authors should clarify whether the responses they received were based on objective data (active surveillance, databases) or subjective opinions of the responders. Line 142: this sentence is confusing. I suggest that Table 2A be omitted. The authors should clarify the percentages of all patients with CKD who had various risk factors (comorbidities, HIV drug toxicity, etc). A Table with this information would be helpful. Presenting the data as done here (i. e. percent of centers that reported various factors as the first, second, and third leading causes of CKD) is hard to follow.
Author Response
Thank you for all the comments and remarks. We addressed all of them but one (please see the explanation below) in the corrected manuscript. To summarise:
- we added the information on response rates
- we were not able to identyfy any pattern of nonresponders influencing the results, (please note that the response rate calculated per country was as high as 80%)
- we added a section in "Methods" explaining the process of data collection
- we rephrased sentences in line 142 to make it more clear
- we insist on showing the graphs because from our point of view, it is crucial to show that the majority of patients has CKD non-related to HIV. It may not be perfect but the easiest way of showing this important message. The numbers regarding the causes of CKD are mentioned in the text.
- we spotted another mistake in numbering the figures and corrected it
Round 2
Reviewer 1 Report
Dear!
All requested changes have been met
Author Response
Thank you. As I understand all necessary changes have been made.